# A Comparative Study of Yeasts for *Rosa roxburghii* Wine Fermentation

**Zhi-Hai Yu** [1],*[iD], **Gui-Dan Huang** [1], **Xiao-Yan Huang** [1], **Jiang-Hua Pu** [1], **Jia-Sheng Wu** [1], **Li-Rong Yue** [1],
**William James Hardie** [2], **Xiao-Zhu Liu** [1] **and Ming-Zheng Huang** [1],*

[1]  College of Food and Pharmaceutical Engineering, Guizhou Institute of Technology, Guiyang 550003, China;
     huang3262022@163.com (G.-D.H.); xiaoyanhuang0630@163.com (X.-Y.H.); puwork2455@163.com (J.-H.P.);
     qgsys2016@163.com (J.-S.W.); ylr120408@163.com (L.-R.Y.); liuxiaozhu_840914@163.com (X.-Z.L.)
[2]  Institute of Urban and Rural Mining, Changzhou University, Changzhou 213164, China; jim@ecovinia.com
*    Correspondence: 123keyan@163.com or 20160739@git.edu.cn (Z.-H.Y.);
     huangmingzheng@git.edu.cn (M.-Z.H.); Tel.: +86-173-8515-5525 (Z.-H.Y.)

**Abstract:** Wine produced by fermentation of Chestnut rose (*Rosa roxburghii*) hips, known as cili (Chinese-Mandarin), in Guizhou province, and other places in China is becoming popular but there is limited knowledge of suitable yeast strains for its production. In this study, we first investigated the oenological properties of six commercial *S. cerevisiae* yeast strains (X16, F33, SH12, GV107, S102, RMS2), one commercial *Saccharomyces cerevisiae* var. *bayanus* (S103), one commercial, non-*Saccharomyces* yeast strain, *Torulaspora delbrueckii* Prelude, and one indigenous *S. cerevisiae* strain, CZ, for cili wine fermentation. We measured the key traits of each of the yeast strains, viz., sulfite resistance, flocculation, hydrogen sulfide production capacity, fermentation rate, and yeast growth curves. Subsequently, we measured the resultant wine characteristics, viz., pH, alcohol content, residual sugar, titratable acidity, volatile acidity, ascorbic acid content and headspace volatile compounds. The overall suitability of each yeast type was evaluated using a multi-factor, unweighted, scorecard. On that basis, RMS2 was the most suitable, and closely followed by CZ and X16. This study is the first comparative evaluation of yeasts for cili wine production and provides a preliminary guide for their selection.

**Keywords:** *Rosa roxburghii* wine; yeast strains; fermentation; winemaking characteristics

## 1. Introduction

Cili, a local name of *Rosa roxburghii* Tratt., in Guizhou province, China, which refers to its spiny, pear-shaped, fruit (hips) (Figure 1), is an underutilized plant of the *Rosaceae* family [1]. While domestication and utilization of cili has attracted the attention of scholars and government in recent years due to its health-promoting properties [2,3], there are very few publications concerning the application of contemporary wine fermentation technologies for the production of 'Western-style' wines from this fruit; either in Chinese or English.

Even though its fruit has a sour, astringent, and bitter taste [4], coupled with low single fruit weight (*ca.* 15 g), short harvest time and a short storage life [5], our previous investigations have found that cili is especially suitable for making fruit wine owing to its high aroma retention and distinctive flavor [5] [Figure 1c].

There are few, if any, commercial wines fermented from cili. Traditional Chinese 'wines' made from cili are made by infusing the fruit in alcohol prepared from other substrates rather than fermentation of the fruit. Furthermore, most commercial yeasts for winemaking have been selected on the basis of their performance in fermenting grapes. Accordingly, there is little information regarding the most suitable yeast for fermenting cili and the most suitable yeast(s) are yet to be determined.

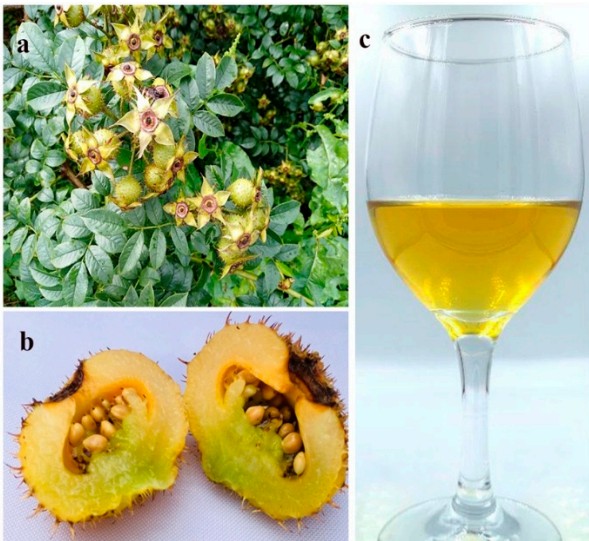

**Figure 1.** Cili, *Rosa roxburghii*, cv. Guinong 5 (**a**) growth habit (**b**) mature fruit and (**c**) wine.

Initially, to determine the most yeast suitable for cili fermentation, our group selected yeasts from spontaneously fermented cili juice and studied their biodiversity coupled with their behaviors during fermentation [6]. Those results provided the direction for subsequent screening, in which six strains that most enhanced the aroma of fermented juice were isolated. We showed that *Wickerhamomyces anomalus* strain C4 had high β-glucosidase activity, which can lead to aroma enhancement [4]. *Hanseniaspora uvarum* strains F13 [7], F119 [8], C26, C31 and C110 [9] all had the ability to reduce the acidity and increase the aroma intensity and complexity of cili wine during laboratory-scale fermentation. One other potentially useful indigenous *S. cerevisiae* yeast, viz., CZ, was also isolated from fermented cili juice [5]. In addition, Liu et al. 2021 [8] used commercial yeast *S. cerevisiae* X16 for cili wine ferments.

Among non-*Saccharomyces* yeasts, *Torulaspora delbrueckii* has attracted considerable interest [10]. It is an osmotolerant yeast, which in grape ferments produces lower concentrations of volatile acidity, and higher concentrations of mannoproteins, which result in a smooth mouth feel and enhanced palate weight [11]. It also generates flavor complexity, due to the production of high concentrations of medium chain fatty-acid esters. It may promote malolactic fermentation, but this aspect may be strain dependent. *T. delbrueckii* is also able to compete well with undesirable yeasts during the early stages of fermentation while the alcohol content is low [10]

Although Ianieva et al., 2020 [12] reported a *T. delbrueckii* strain with high β-glucosidase activity, this appears to be a strain-dependent trait, see Comitini et al., 2011 [13], and Escribano et al., 2017 [14].

Sequential inoculation with *T. delbrueckii* followed by *S. cerevisiae* is now commonly advocated for imparting its desirable attributes to grape wines [15–17].

Consequently, in this report, we describe a comparative study of the oenological properties of yeasts with a wide range of attributes including tolerance of fermentation conditions, low production of off-flavors as well as development of desirable sensory qualities, and from diverse sources including seven *S. cerevisiae* strains (six commercial and one indigenous strain), one commercial *S. cerevisiae* var. *bayanus* and one commercial non-*Saccharomyces* yeast, *Torulaspora delbrueckii*, to determine the most suitable yeasts for making cili wine.

## 2. Materials and Methods

### 2.1. Materials

2.1.1. Chemicals and Yeast Strains

Analytical grade reagents, solvents, and standards used here were purchased from a local supplier (Reggie Biology) in Guiyang, China. Winemaking materials, viz., tartaric acid, potassium bicarbonate, potassium metabisulfite and pectinase (Lafase He Grand Cru) were sourced from Laffort®, Bordeaux, France. The sources of yeast for comparative fermentations are presented in Table 1.

**Table 1.** Source of yeast strains.

| | Strain | Source |
|---|---|---|
| *Saccharomyces cerevisiae* | X16 F33 RMS2 | Laffort, France |
| | SafOEno SH 12 SafOEno GV107 SafOEno CK S102 | Fermentis, France |
| | CZ * | Guizhou Institute of Technology |
| *S. cerevisiae* var. *bayanus* | SafOEno BC S103 | Fermentis, France |
| *Torulaspora delbrueckii* | Viniflora Prelude | Chr. Hansen, Denmark |

* Strain CZ was previously isolated from a spontaneous fermentation in cili juice [3].

2.1.2. Preparation of Cili Juice

Cili, cv. Guinong 5, juice was provided by Guizhou Hongcai Investment Group Co., Ltd. The juice had been extracted by belt press, pasteurized by ultra-high temperature (UHT) processing at 125 °C for 10 s, then directly poured into a 200-liter sterilized aluminum foil bag with low oxygen permeability, which was sealed and stored in a closed, steel drum at room temperature for six months.

The juice was prepared for fermentation according to treatments shown in Table 2. After those treatments, the juice was high temperature, short time (HTST) pasteurized at 65 °C for 15 min. The prepared juice was used for winemaking and other tests.

**Table 2.** Treatments of cili juice.

| Parameter | Target Value | Method |
|---|---|---|
| Juice density (Brix) | 24 | Add sucrose |
| YAN * (mg/L) | ~140 | Add Thiazote® (Laffort) |
| pH | 3.6 | Add tartaric acid/Potassium bicarbonate |
| Dilution ratio) | 2 | Add water |
| Suspended solids (NTU) | <100 or 1–2% | Centrifugation |

* Yeast assimilable nitrogen (YAN) was quantified as free amino nitrogen (FAN) (K-PANOPA, and ammonium ion (Ammonia-Rapid) kits, respectively (Megazyme, Wicklow, Ireland). Samples were centrifuged at $1096\times g$ for 5 min prior to analysis [18].

2.1.3. Preparation of Cili Wine Using Different Yeast Strains

Aliquots of cili juice (1.5 L) were fermented in 2 L flasks fitted with water-filled air locks. The juice was inoculated with a fresh culture of a single yeast strain to obtain approximately $1 \times 10^6$ CFU/mL starting concentration. For the mixed inoculation, the juice was first inoculated with Prelude, followed by CZ five days later. The initial concentration of each of those strains at inoculation was $5 \times 10^5$ CFU/mL, i.e., a combined concentration of $1 \times 10^6$ CFU/mL. After inoculation, the flasks were placed in a cool room (18 °C) for fermentation. After the alcoholic fermentation (around 15 days), the wine was racked from fermentation lees and potassium metabisulfite was added to provide 50 mg/L total

$SO_2$. The wine was settled at 4°C. During settling, the wine was racked from the residue four times. After about six months, the finished wine was chemically analyzed.

*2.2. Methods*

2.2.1. Sulfite Tolerance and Flocculation

Sulfite tolerance and flocculation of each strain were determined in a 50 mL glass tube containing 40 mL of the prepared cili juice. The yeast concentration in the juice following inoculation was approximately $1 \times 10^6$ CFU/mL. After inoculation, each tube was cultured at 18 °C for 72 h. Then, the spectral absorption of each yeast solution was determined at 600 nm (A600) by ultraviolet spectrophotometer (SOPTOP, UV2400, China).

Sulfite tolerance of each yeast was evaluated in the prepared cili juice with 0, 100, 200, 300, and 400 mg/L total $SO_2$ (by $K_2S_2O_5$ addition) based on the method of Porter et al. 2019 [19]. The relative biomass concentration of each strain compared with the control without sulfur addition was used as an index of sulfite tolerance. Biomass concentration was inferred from absorption of well shaken cultures and relative concentrations were calculated according to the following formula.

$$\text{Relative biomass concentration (\%)} = \text{A600n/A600a} \times 100 \tag{1}$$

where A600n is the absorbance of the culture, and A600a is the absorbance of the culture without $SO_2$ addition.

Flocculation was tested according to the protocol of Vigentini et al. (2017) [20] with minor modification. Briefly, the A600 values were measured immediately, and after four hours during which the cuvette was kept undisturbed at room temperature. The degree of flocculation (*F*) was calculated as follows:

$$\text{Flocculation} = \text{A600 after 4 h/A600 at inoculation} \tag{2}$$

with values ranging from 0–30 (very flocculent), 31–70 (moderately flocculent) and 71–100 (poorly flocculent).

2.2.2. Hydrogen Sulfide Production Capacity

The hydrogen sulfide production capacity of the yeasts was determined according to the method of Vigentini et al. (2017) [20]. Briefly, 5 µL of a fresh culture was spotted on BIGGY agar plates. After inoculation, the plates were incubated at 28°C for 96 h. The color of the colonies ranged from white/cream to brown/black as a function of increasing hydrogen sulfide production. The number of symbols '+' was used to record the degree of coloration.

2.2.3. Fermentation Vigor and Growth Curve Determination

The growth curve of each yeast and fermentation vigor were recorded. The cell growth curve of each yeast was plotted from spectral absorbance measured at 600 nm every 24 h over 15 consecutive days. The time (days) to reach the stationary phase was used as an indicator of fermentation efficiency. Fermentation vigor was determined as grams of $CO_2$ lost within 48 h of inoculation.

2.2.4. Chemical Analysis of Resultant Wines

Cili wine resulting from each ferment was analyzed for pH, ethanol concentration (%vol), residual sugar expressed as glucose (g/L), titratable acidity (g/L) expressed as tartaric acid, volatile acidity (g/L) expressed as acetic acid, and ascorbic acid content (g/L) according to the standard protocols of the International Office of the Vine (OIV), (2019) [21].

2.2.5. Volatile Components of Resultant Wines

After solid-phase microextraction (SPME), the volatile compounds of the wine headspace were analyzed by GC-MS analysis according to the method of Zhao et al. (2021) [22]. The

volatiles were identified by comparison of their *RI*s and experimental mass spectra with those published in the literature and NIST Mass Spectral libraries. Semi-quantification of the volatiles was conducted using cyclohexanone as the internal standard according to the method of Huang et al. 2022 [23] using the formula: $\varrho i = Ai/As \times \varrho s$, where As was the peak area of the internal standard; Ai was the peak area of the unknown compound, $\varrho s$ was the mass concentration (mg/L) of the internal standard, and $\varrho i$ was the mass concentration (mg/L) of the unknown compound.

Odor activity values (OAVs) of the wine headspace volatile compounds were determined as the ratio of the concentration of each compound to its detection threshold in water or wine/ethanol solution containing less than or equal to 14% vol alcohol. Threshold values were taken from the literature.

### 2.3. Suitability of Yeast Strains for Cili Winemaking

The overall suitability of each yeast strain for cili winemaking was determined from a multi-factor scorecard based on the analytical data. In the absence of further information regarding their relative impacts, individual parameters received no differential weighting (Table 3).

**Table 3.** Scoring standards for yeast traits for winemaking and the resultant wine attributes.

| Items | Range | | |
|---|---|---|---|
| **Yeast Traits for Winemaking** | | | |
| SO$_2$ tolerance (Relative biomass concentration %) | 60–70 | 71–80 | >80 |
| Score | 1 | 2 | 3 |
| Flocculation (*F* value) | 71–100 | 31–70 | 0–30 |
| Score | 1 | 2 | 3 |
| H$_2$S production capacity | $\geq$3 | 1–2 | 0 |
| Score | 1 | 2 | 3 |
| Growth curve (day) | >8–14 | >6–7 | >0–5 |
| Score | 1 | 2 | 3 |
| Fermentation vigor (g CO$_2$/100 mL) | 0–0.60 | 0.61–1.20 | 1.21–1.80 |
| Score | 1 | 2 | 3 |
| **Wine Attributes** | | | |
| pH | 2.0–3.0 | 3.1–4.0 | 4.1–5.0 |
| Score | 1 | 3 | 1 |
| Volatile acidity (g/L) | 0–0.6 | 0.61–1.2 | >1.2 |
| Score | 3 | 2 | 0 |
| Titratable acidity (g/L) | 0–6 | 7–12 | 13–18 |
| Score | 3 | 2 | 1 |
| Residual sugar (g/L) | $\leq$4 | 4.1–12 | $\geq$12.1 |
| Score | 3 | 2 | 1 |
| Alcohol (% vol) | 7.0–9.0 | 9.1–11.0 | 11.1–13.0 |
| Score | 1 | 2 | 3 |
| Ascorbic acid (g/L) | 1–2 | 3–4 | 5–6 |
| Score | 1 | 2 | 3 |
| *Volatile Components* Higher Alcohols (mg/L) | 100–130 | 131–160 | >160 |
| Score | 3 | 2 | 1 |
| Acids (mg/L) | 20–35 | 36–45 | >45 |
| Score | 3 | 2 | 1 |
| Esters (mg/L) | 50–100 | 100–200 | >200 |
| Score | 2 | 3 | 1 |
| Aldehydes (mg/L) | >1 | 0.1-1.0 | <0.1 |
| Score | 3 | 2 | 1 |
| Phenols (mg/L) | 0.1–5 | 5.1–10 | >10 |
| Score | 1 | 2 | 3 |

### 2.4. Statistical Analysis

All experiments were performed in triplicate, and the data were expressed as the mean ± standard deviation. SPSS software (IBM Corp., version 23.0, Armonk, NY, USA) was used for statistical analysis. Significant differences among the samples were calculated using one-way ANOVA at the 5% level ($p < 0.05$). GraphPad Prism 8.0.1 (San Diego, CA, USA) was used to generate the figures.

## 3. Results

### 3.1. Traits of Yeast Strains for Winemaking

#### 3.1.1. Tolerance of Total $SO_2$

After 72 h of culture without $SO_2$ addition, the absorbance of *Torulaspora delbrueckii* Prelude, (1.06), was lower than that of all the *Saccharomyces* strains, (1.41–1.49), (Table 4). In terms of $SO_2$ tolerance, the relative biomass concentration of all strains decreased as the total $SO_2$ concentration increased. There were significant differences in relative biomass concentration among the various yeast strains. Interestingly, the non-commercial *Saccharomyces* strain CZ was amongst those with significantly higher tolerance. Notably, at the highest total $SO_2$ concentration, the tolerance of the *Torulaspora delbrueckii* strain was significantly greater than all the *Saccharomyces* strains.

**Table 4.** $SO_2$ tolerance of yeast strains.

| Group | Strains | Control * | Relative Biomass Concentration (%) Total $SO_2$ mg/L | | | |
| --- | --- | --- | --- | --- | --- | --- |
| | | | 100 | 200 | 300 | 400 |
| *Saccharomyces cerevisiae* | X16 | 1.45 ± 0.01 | 85.69 ± 1.46bc | 56.59 ± 0.71a | 52.94 ± 0.42c | 41.38 ± 0.38b |
| | S102 | 1.47 ± 0.01 | 78.70 ± 3.47d | 56.28 ± 0.60a | 52.90 ± 0.92c | 40.85 ± 0.49b |
| | F33 | 1.42 ± 0.02 | 66.79 ± 19.39e | 62.62 ± 18.51c | 59.10 ± 0.26c | 43.43 ± 1.71b |
| | GV107 | 1.41 ± 0.01 | 91.71 ± 0.86abc | 59.04 ± 0.67a | 54.28 ± 0.85c | 41.79 ± 1.29b |
| | CZ | 1.47 ± 0.02 | 90.64 ± 4.01a | 57.70 ± 1.21a | 52.95 ± 0.49c | 41.67 ± 0.90b |
| | RMS2 | 1.43 ± 0.01 | 85.91 ± 0.75cd | 83.83 ± 0.34b | 76.85 ± 1.19a | 42.86 ± 0.37b |
| | SH12 | 1.47 ± 0.02 | 89.16 ± 1.29ab | 56.08 ± 1.31a | 52.84 ± 0.64c | 41.51 ± 1.19b |
| *S. cerevisiae* var. *bayanus* | S103 | 1.49 ± 0.01 | 87.89 ± 1.03ab | 51.19 ± 2.74a | 50.70 ± 0.73c | 40.91 ± 0.36b |
| *Torulaspora* | Prelude | 1.06 ± 0.01 | 77.50 ± 1.37d | 73.28 ± 1.21b | 64.86 ± 1.42b | 60.75 ± 1.83a |

* This column shows A600 of each strain without $SO_2$ addition. Values in the same column with different lowercase letters are significantly different ($p \leq 0.05$).

#### 3.1.2. Flocculation

The flocculation value (*F*) of all strains was less than 30 (Figure 2), which indicated that all tested strains were very flocculant.

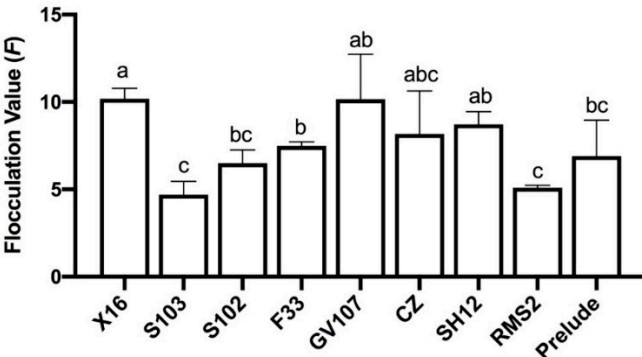

**Figure 2.** Flocculation of nine yeast strains. Different lowercase letters above the standard deviation bar indicate significant difference ($p \leq 0.05$).

### 3.1.3. H$_2$S Production Capacity

After 96 h incubation, the colony color of strains F33 and Prelude were deep brown, S103 was brown, X16, GV107, and S102 were light brown, but CZ, RMS2, and SH12 were white cream (Figure 3). Those results indicated that CZ, RMS2, and SH12 were the strains with the weakest capacity to produce hydrogen sulfide.

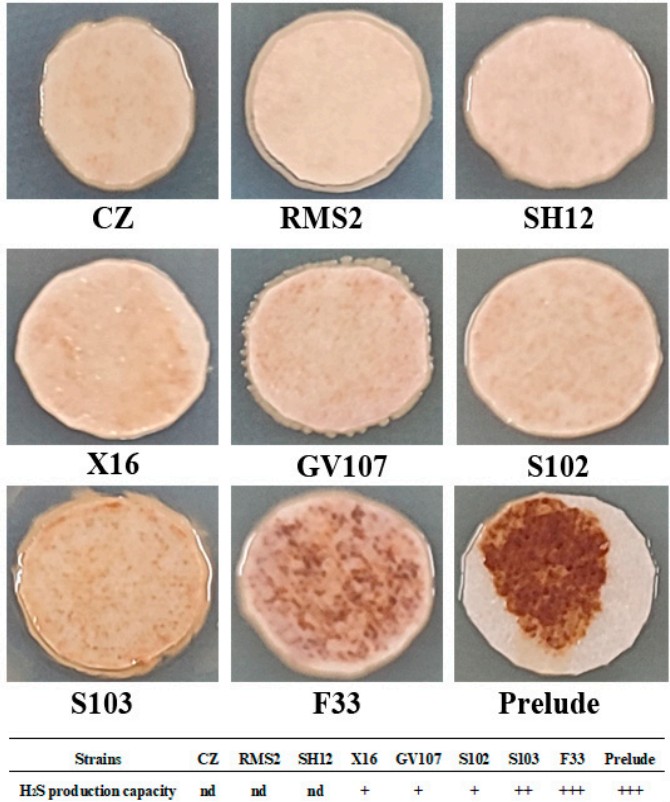

| Strains | CZ | RMS2 | SH12 | X16 | GV107 | S102 | S103 | F33 | Prelude |
|---|---|---|---|---|---|---|---|---|---|
| H$_2$S production capacity | nd | nd | nd | + | + | + | ++ | +++ | +++ |

**Figure 3.** H$_2$S production capacity of yeast strains. In the table, the greater number of + indicated the stronger the H$_2$S production capacity, and nd indicates no obvious H$_2$S detection.

### 3.1.4. Growth Curve and Fermentation Vigor

During alcoholic fermentation, yeast cell growth and fermentation vigor were monitored. Except for Prelude and Pre/CZ, there was no obvious lag phase following inoculation (Figure 4a). The most rapid growth of all strains took place during the first six days of fermentation. The stationary phase of growth of all yeast strains, except for Prelude, was reached after eight days of fermentation.

The fermentation vigor of the yeast strains ranged from about 0.03 to 1.70 g CO$_2$/100 mL. Strain F33 (*ca.* 1.7g CO$_2$/100 mL) had the strongest fermentation vigor followed in decreasing order by S103, S102, RMS2, GV107, X16, CZ, SH12, Pre-CZ, and Prelude, with CO$_2$ generations of 1.53, 1.50, 1.46, 1.39, 1.31 1.31, 1.10, 0.20, and 0.03 g CO$_2$/100 mL, respectively (Figure 4b).

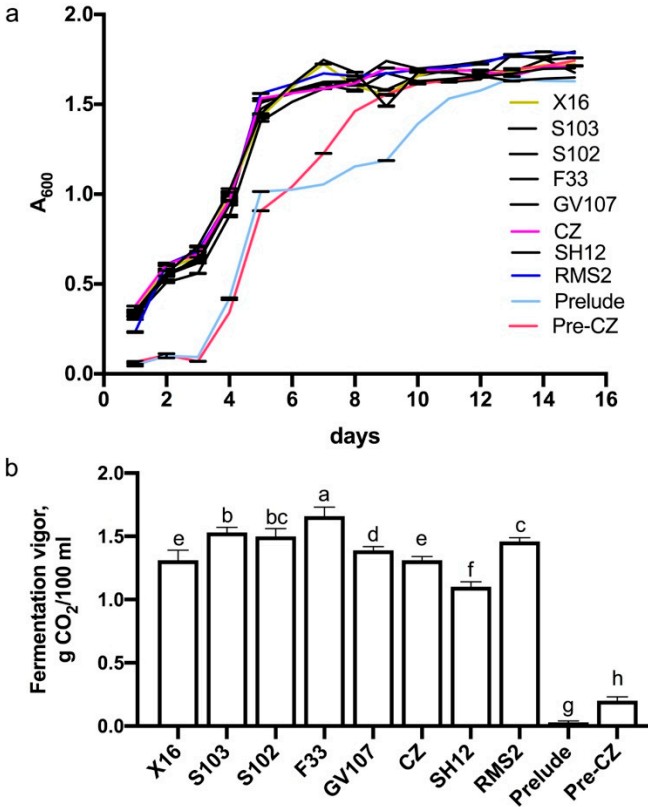

**Figure 4.** Growth curves (**a**) and fermentation vigor (**b**) of yeast strains. Different lower letters in (**b**) above the standard deviation bar indicate significant difference ($p \leq 0.05$).

*3.2. Wine Analysis*

3.2.1. Chemical Components

As shown in Table 5, the pH of cili wines fermented by the various yeast strains ranged from 3.64 to 3.71. The volatile acidity of all wines was in the range of 0.34 g/L (F33) to 0.71 g/L (S102). The volatile acidity of the wine fermented with S102 was significantly higher than that produced by the other strains apart from GV107 (0.65 g/L). Titratable acidity ranged from 10.70 g/L (F33) to 11.68 g/L (CZ), and there were no significant differences between strains. Residual sugar ranged from 58.19 g/L (RMS2) to 111.16 g/L (Prelude), which indicated that among the strains, fermentation was inhibited to various degrees. Alcohol content ranged from 7.58 %vol (Prelude) to 11.55%vol (RMS2). The alcohol content of RMS2 was significantly higher than that of all other strains except for X16 (10.50 %vol) and S103 (10.50 %vol). Ascorbic acid contents ranged from 5.35 g/L (F33) to 5.68 g/L (Prelude). The ascorbic acid content of the wine fermented with Prelude was significantly higher than that of RMS2 (5.37 g/L) and S102 (5.36 g/L).

**Table 5.** Chemical components of cili wines fermented with different yeast strains.

| Wine | pH | Volatile Acidity g/L | Titratable Acidity g/L | Residual Sugar g//L | Alcohol Concentration %vol | Ascorbic Acid g/L |
|---|---|---|---|---|---|---|
| X16 | 3.65 ± 0.02bc | 0.56 ± 0.08bc | 11.03 ± 0.37 | 82.32 ± 9.26bcd | 10.50 ± 0.55ab | 5.63 ± 0.13ab |
| S103 | 3.64 ± 0.01c | 0.61 ± 0.04bc | 11.46 ± 0.09 | 74.56 ± 1.32d | 10.50 ± 0.55ab | 5.45 ± 0.22ab |
| S102 | 3.65 ± 0.01bc | 0.71 ± 0.10a | 11.54 ± 0.65 | 85.75 ± 1.66c | 9.08 ± 1.02b | 5.36 ± 0.10b |
| F33 | 3.65 ± 0.03bc | 0.34 ± 0.05d | 10.70 ± 0.55 | 75.57 ± 3.71d | 9.43 ± 0.48b | 5.35 ± 0.29ab |
| GV107 | 3.71 ± 0.01a | 0.65 ± 0.06ab | 11.37 ±0.33 | 65.97 ± 6.13de | 10.00 ± 0.02b | 5.60 ± 0.13ab |
| CZ | 3.65 ± 0.04bc | 0.63 ± 0.04b | 11.68 ± 0.31 | 81.51 ± 5.09cd | 9.95 ± 0.61b | 5.55 ± 0.49ab |
| SH12 | 3.66 ± 0.01b | 0.55 ± 0.06c | 11.52 ± 0.25 | 85.78 ± 5.46bcd | 9.13 ± 0.29b | 5.58 ± 0.41ab |
| RMS2 | 3.68 ± 0.02ab | 0.52 ± 0.06c | 11.28 ± 0.14 | 57.47 ± 2.71e | 11.55 ± 0.50a | 5.37 ± 0.09b |
| Prelude | 3.64 ± 0.08abc | 0.56 ± 0.07bc | 11.32 ± 0.72 | 107.45 ± 48.28a | 7.58 ± 3.22c | 5.68 ± 0.14a |
| Pre-CZ | 3.66 ± 0.01b | 0.56 ± 0.08bc | 11.20 ± 0.15 | 91.82 ± 2.60b | 8.78 ± 0.47bc | 5.57 ± 0.30ab |

Values in the same column with different lowercase letters are significantly different ($p \leq 0.05$).

3.2.2. Headspace Volatile Compounds of Cili Wines

Sixty-one volatile compounds detected in the headspace of cili wines fermented by the different yeast strains are presented in Table S1. They comprised 16 higher alcohols, five acids, 26 esters, two aldehydes, eight volatile phenols, and four others. The number of volatile compounds in the X16, S103, S102, F33, GV107, CZ, SH12, RMS2, Prelude and Pre-CZ wines were 28, 26, 29, 33, 31, 31, 35, 29, 47 and 34, respectively. Alcohols and esters accounted for the highest proportions of the six chemical groups in each of the samples (Table S1). The percentage of higher alcohols and esters ranged from 21.74 to 33.33 and 39.39 to 52.17%, respectively. Higher alcohols and esters together accounted for more than 60% of the total volatiles in each wine.

Higher Alcohols

Sixteen higher alcohols were identified in the headspace of the cili wines (Table S1). Higher alcohol content ranged from 120.73 to 177.23 mg/L in the SH12 and GV107 wine headspaces, respectively. Each cili wine headspace contained 2-methyl-1-butanol, 1-pentanol, hexyl alcohol, leaf alcohol, (R, R)-2,3-butanediol, and phenethyl alcohol, and those alcohols were the main components of higher alcohols in all cili wine headspace samples. The higher alcohol with the highest headspace concentration was 3-methyl-1-butanol, which represented 68.15 and 74.16% of all alcohols in the S103 and X16 wine headspaces, respectively. Other alcohols were only found in the headspace of one or several wines and the content of most was less than 20 mg/L, except for leaf alcohol content, which ranged from 16.60 to 27.56 mg/L.

Acids

A total of five acids were detected in the headspace of the wines and their total concentration ranked third among the six volatile chemical groups (Table S1). Acid content ranged from 22.58 to 54.00 mg/L in the GV107 and Prelude wine headspaces, respectively. Acetic acid, octanoic acid, and decanoic acid were detected in the headspace of all wines. Octanoic acid was the acid with the highest content, accounting for about 65.75 (F33)-83.14% (S103) of the total volatile acids content. (E)-3-hexenoic acid was only detected in the Prelude wine headspace and 9-decenoic acid was not found in the headspace of the S103, F33 and GV107 wines.

Esters

Twenty-six esters were detected in the cili wine headspace (Table S1). The highest number of esters, viz., 24, were identified in the headspace of Prelude wine, followed by 16 (F33, CZ and SH12 wines), 14 (S102 and RMS2 wines), 13 (X16, GV107 and Pre-CZ wines), and 12 (S103 wine). The total headspace concentration of esters ranged from 603.60 mg/L (Prelude) to 182.64 mg/L (S103). The esters detected in the headspace of all ten wines were ethyl acetate, ethyl butyrate, isoamyl acetate, ethyl hexanoate, hexyl acetate, (E)-3-hexenyl acetate, ethyl octanoate, ethyl decanoate, ethyl 9-decenoate, and phenethyl acetate. The ester with the highest concentration in the ten wines was ethyl octanoate, which represented 31.14 and 53.35% of the total ester content in the S103 and RMS2 wines, respectively.

Aldehydes

Two aldehydes, acetaldehyde and 2,4-dimethyl benzaldehyde were detected in the cili wine headspaces (Table S1).

Acetaldehyde was detected in all wine headspaces, while 2,4-dimethyl benzaldehyde was not found in the headspace of GV107, CZ and RMS2 wines. The total aldehyde concentration in each wine headspace was less than 2 mg/L, except for that of Pre-CZ and CZ wine, in which the total concentration was 4.03 and 2.14 mg/L, respectively.

Volatile Phenols

There were eight volatile phenols detected in the wine headspaces (Table S1). Butylate hydroxytoluene were detected in the headspace of each wine, except for that of GV107. Other volatile phenols were only found in one or more cili wine headspaces and the content of most was less than 2 mg/L.

Other Volatile Compounds

Four compounds, classified as 'Others', were detected at very low concentrations in some wine headspace samples (Table S1).

3.2.3. Aroma-Contributing Compounds of Cili Wine

Although a total of 61 volatile compounds were detected in the headspaces of the cili wines, not all are likely to have contributed to the overall aroma and flavor of the wine. Appreciable contributions depend on the concentration of the compound in the wine and its odor detection threshold. Twenty-eight of the volatile compounds we detected in the cili wines had OAVs $\geq 1$ (Table 6). They comprised ten higher alcohols, three fatty acids, twelve esters and three phenols. Both their presence and their OAVs varied according to yeast strain. The number of those compounds in Prelude wine was 25, followed by 19 (S102, F33, RMS2, and Pre-CZ wines), 18 (GV107 and CZ wines), 17 (SH12 wine), 16(S103 wine) and 15 (X16 wine). Fourteen of those compounds, viz., 3-methyl-1-butanol, 1-pentanol, leaf alcohol, (R, R)-2,3-butanediol, octanoic acid, decanoic acid, ethyl butyrate, isoamyl acetate, ethyl hexanoate, hexyl acetate, (E)-3-hexenyl acetate, ethyl octanoate, ethyl decanoate, and phenethyl acetate, were found in each wine. Some odor active compounds were only detected in a single wine, viz., 1-penten-3-ol (green, fruity) and 3-ethoxy-1-propanol (fruity) in Prelude wine, and decyl alcohol (fatty, sweet) in Pre-CZ wine.

Esters were the dominant aroma active volatiles in the wines. Within that group, the acetate ester, isoamyl acetate (sweet, fruity, banana), contributed appreciably to the aroma of each of the wines. However, except for isoamyl acetate, the magnitude and range of OAVs of ethyl esters of MCFAs, viz., ethyl hexanoate (sweet fruity, pineapple), ethyl octanoate (waxy, fruity, winey), ethyl decanoate (sweet, fruity, apple) were greater, by two–three orders of magnitude, than other volatiles detected in the wines. For each of the wines, the OAV of ethyl hexanoate (sweet, fruity and pineapple) greatly exceeded that of all others but also varied greatly according to the yeast strain that generated it. The dominant contribution of ethyl hexanoate to the aroma of each wine accords with its similar dominance in cili juices [23]. The OAVs of ethyl hexanoate and isoamyl acetate were highest in wines fermented by *T. delbrueckii*, Prelude.

*3.3. Scoring of Oenological Properties of Yeast Strains for Cili Fermentation and the Resultant Wine*

Based on the scores derived from the analytical results and assigned to each yeast strain and the resultant wines (Table 7), the *Saccharomyces* strains RMS2 was the most suitable for cili winemaking. Among the other yeasts, strains CZ and X16, scored most highly. Although the yeast attributes of the Pre/CZ combination were not scored, it is noted that even had it received the maximum score for those attributes, the total score for that combination was lower than the more suitable strains.

**Table 6.** Odor activity values of headspace volatile compounds of cili wines fermented by different yeasts.

| No. | Compounds | CAS | Odor Quality | Odor Threshold (mg/L) | X16 | S103 | S102 | F33 | GV107 | SH12 | RMS2 | CZ | Prelude | Pre-CZ |
|---|---|---|---|---|---|---|---|---|---|---|---|---|---|---|
| | | | | | | | | | OAV | | | | | |
| | *Higher alcohols* | | | | | | | | | | | | | | |
| 1 | 1-Propanol | 71-23-8 | Fusel, alcoholic | 314 [24] | <1 | nd | <1 | nd | nd | nd | <1 | <1 | nd | <1 |
| 2 | 2-Methyl-1-propanol | 78-83-1 | fusel whiskey | 82 [24] | <1 | 1 | <1 | <1 | <1 | <1 | <1 | <1 | <1 | <1 |
| 3 | 1-Penten-3-ol | 616-25-1 | Green, fruity | 0.4 [23] | nd | nd | nd | nd | nd | nd | nd | nd | 28 | nd |
| 4 | 1-Butanol | 71-36-3 | Fusel oil sweet balsam whiskey | 160 [24] | nd | nd | nd | <1 | <1 | nd | nd | nd | nd | nd |
| 5 | 3-Methyl-1-butanol | 123-51-3 | Alcoholic, fruity | 40 [25] | 3 | 3 | 2 | 2 | 2 | 2 | 2 | 2 | 2 | 2 |
| 6 | 1-Pentanol | 71-41-0 | Fermented, bready, fusel | 0.1502 [26] | 2 | 2 | 2 | 1 | 2 | 3 | 2 | 1 | 2 | 2 |
| 7 | Hexyl alcohol | 111-27-3 | Pungent, fruity, alcoholic | 8 [25] | <1 | <1 | <1 | <1 | <1 | <1 | <1 | <1 | <1 | <1 |
| 8 | (E)-3-Hexen-1-ol | 928-97-2 | Green | 1 [24] | nd | nd | nd | nd | <1 | nd | nd | nd | nd | <1 |
| 9 | 3-Ethoxy-1-propanol | 111-35-3 | Fruity | 0.1 [24] | nd | nd | nd | nd | nd | nd | nd | nd | 2 | nd |
| 10 | Leaf alcohol | 928-96-1 | Green, grassy | 1 [24] | 27 | 26 | 26 | 17 | 28 | 25 | 26 | 24 | 22 | 27 |
| 11 | (R, R)-2,3-Butanediol | 24347-58-8 | Buttery, Creamy, Fruity | 0.0951 * [26] | 26 | 17 | 21 | 32 | 31 | 21 | 30 | 20 | 27 | 22 |
| 12 | 2-Nonanol | 628-99-9 | Waxy, green, Creamy | 0.075 [27] | nd | nd | nd | nd | nd | 7 | nd | nd | nd | nd |
| 13 | 1-Octanol | 111-87-5 | Waxy, green, fatty | 10 [24] | <1 | nd | nd | nd | nd | nd | nd | <1 | <1 | <1 |
| 14 | Furfuryl alcohol | 98-00-0 | Musty, sweet | 15 [28,29] | nd | nd | nd | nd | nd | <1 | nd | nd | nd | nd |
| 15 | Phenethyl alcohol | 60-12-8 | Sweet, floral | 14 [25] | nd | <1 | 1 | 2 | 1 | <1 | 1 | <1 | 1 | <1 |
| 16 | Decyl alcohol | 112-30-1 | Fatty, sweet | 0.5 [24] | nd | nd | nd | nd | nd | nd | nd | nd | nd | 2 |
| | *Acids* | | | | | | | | | | | | | | |
| 17 | Acetic acid | 64-19-7 | Vinegar | 300 [30] | <1 | <1 | <1 | <1 | <1 | <1 | <1 | <1 | <1 | <1 |
| 18 | Octanoic acid | 124-07-2 | Vegetable, cheesy | 10 [25] | 2 | 2 | 2 | 3 | 2 | 2 | 2 | 3 | 4 | 4 |
| 19 | Decanoic acid | 334-48-5 | Sour, fatty | 0.5 [24] | 2 | 2 | 5 | 34 | 5 | 6 | 4 | 3 | 15 | 8 |
| 20 | 9-Decenoic acid | 14436-32-9 | Waxy, green, fatty | 4.3 [31] | <1 | nd | <1 | nd | nd | <1 | <1 | <1 | 1 | <1 |
| | *Esters* | | | | | | | | | | | | | | |
| 21 | Ethyl acetate | 141-78-6 | Fruity, sweet | 15 [24] | <1 | <1 | <1 | <1 | <1 | <1 | <1 | <1 | 2 | 1 |
| 22 | Isobutyl acetate | 110-19-0 | Sweet, fruity | 3.4 [32] | <1 | nd | nd | <1 | nd | nd | nd | <1 | <1 | <1 |
| 23 | Ethyl propionate | 105-37-3 | Sweet, fruity | 2.1 [24] | nd | nd | nd | nd | nd | nd | nd | nd | <1 | nd |
| 24 | n-Propyl acetate | 109-60-4 | Fusel, sweet, fruity | 65 [28] | nd | nd | nd | nd | nd | nd | nd | nd | <1 | nd |
| 25 | Ethyl butyrate | 105-54-4 | Sweet, fruity | 0.6 [24] | 2 | 3 | 3 | 2 | 3 | 4 | 3 | 3 | 13 | 4 |
| 26 | Isoamyl acetate | 123-92-2 | Sweet, fruity, banana | 0.03 [33] | 804 | 1071 | 626 | 582 | 643 | 614 | 502 | 860 | 2603 | 1300 |
| 27 | Ethyl hexanoate | 123-66-0 | Sweet, fruity, pineapple | 0.014 [25] | 1704 | 1748 | 2729 | 2597 | 2950 | 3118 | 3652 | 3303 | 4993 | 4575 |
| 28 | Hexyl acetate | 142-92-7 | Fruity, green | 1 [29] | 3 | 3 | 3 | 2 | 3 | 4 | 3 | 6 | 10 | 8 |
| 29 | (E)-3-Hexenyl acetate | 3681-82-1 | Sharp fruity, green | 0.87 [34] | 31 | 31 | 24 | 18 | 24 | 27 | 23 | 34 | 58 | 52 |
| 30 | Ethyl octanoate | 106-32-1 | Sweet, musty, fruity | 0.6 [24] | 125 | 95 | 218 | 266 | 210 | 205 | 348 | 253 | 343 | 312 |
| 31 | Ethyl decanoate | 110-38-3 | Sweet, fruity, apple | 0.2 [25] | 25 | 24 | 89 | 329 | 68 | 85 | 126 | 89 | 173 | 97 |
| 32 | 3-methylbutyl octanoate | 2035-99-6 | Sweet, fruity, green | 0.125 [30] | 7 | 5 | 3 | 5 | 4 | 5 | 6 | 6 | 15 | nd |
| 33 | Phenethyl acetate | 103-45-7 | Sweet, honey, floral rosy | 0.25 [33] | 15 | 13 | 19 | 28 | 10 | 7 | 13 | 11 | 32 | 17 |
| 34 | Isoamyl decanoate | 2306-91-4 | Waxy, banana fruity | >5.0 [32] | nd | nd | nd | <1 | nd | nd | <1 | <1 | <1 | nd |
| 35 | Ethyl myristate | 124-06-1 | Sweet, waxy | 0.5 [24] | nd | <1 | 3 | 2 | 1 | 1 | 1 | 2 | 2 | nd |
| 36 | Ethyl laurate | 106-33-2 | Sweet, waxy, floral nuance | 0.5 [24] | nd | nd | 23 | 42 | 18 | nd | 22 | 26 | 42 | 38 |
| 37 | Ethyl palmitate | 628-97-7 | Waxy, fruity, creamy | 1 [24] | nd | nd | nd | <1 | nd | nd | nd | nd | nd | nd |
| | *Aldehydes* | | | | | | | | | | | | | | |
| 38 | Acetaldehyde | 75-07-0 | Pungent, fresh, fruity, musty | 110 [24] | <1 | <1 | <1 | <1 | <1 | <1 | <1 | <1 | <1 | <1 |
| | *Phenols* | | | | | | | | | | | | | | |
| 39 | Butylated hydroxytoluene | 128-37-0 | Mild phenolic camphor | 1 * [35] | <1 | <1 | 2 | <1 | nd | <1 | 18 | 12 | 1 | 20 |
| 40 | Naphthalene | 91-20-3 | Pungent dry tarry | 0.006 * [36] | nd | nd | nd | nd | nd | nd | nd | nd | 13 | nd |
| 41 | Methyl eugenol | 93-15-2 | Spicy, musty, vegetative | 10 [37] | nd | nd | nd | nd | nd | nd | nd | nd | nd | <1 |
| 42 | 2-Ethoxynaphthalene | 93-18-5 | Powder, floral | 0.1 [24] | nd | nd | nd | 7 | nd | nd | nd | nd | 4 | 11 |

Note: Except that the one marked with * refers to the threshold value of the substance in water, the rest refers to which in wine/ethanol solution containing less than or equal to e 14% vol alcohol. See their respective references for details. nd means not detected.

**Table 7.** Score of winemaking traits of various yeast strains and resultant wines.

| | Yeast Attributes | | | | | | | | | | Wine Attributes | | | | | | |
| | Standard Parameters | | | | | | | | | | Headspace Volatiles | | | | | | |
| Yeast Strains | Sulfite Tolerance | Flocculation | H$_2$S Production Capacity | Growth Curve | Fermentation Vigor | pH | Volatile Acidity | Titratable Acidity | Residual Sugar | Ethanol | Ascorbic Acid | Higher Alcohols | Acids | Esters | Aldehydes | Phenols | Total Score |
|---|---|---|---|---|---|---|---|---|---|---|---|---|---|---|---|---|---|
| X16 | 3 | 3 | 2 | 3 | 3 | 3 | 3 | 2 | 1 | 2 | 3 | 2 | 3 | 3 | 3 | 1 | 40 |
| S103 | 3 | 3 | 2 | 3 | 3 | 3 | 2 | 2 | 1 | 2 | 3 | 1 | 3 | 3 | 3 | 1 | 38 |
| S102 | 2 | 3 | 2 | 3 | 3 | 3 | 2 | 2 | 1 | 2 | 3 | 2 | 3 | 1 | 3 | 1 | 36 |
| F33 | 1 | 3 | 1 | 3 | 3 | 3 | 3 | 2 | 1 | 2 | 3 | 3 | 1 | 1 | 3 | 1 | 34 |
| GV107 | 3 | 3 | 2 | 3 | 3 | 3 | 2 | 2 | 1 | 2 | 3 | 2 | 3 | 1 | 3 | 1 | 37 |
| CZ | 3 | 3 | 3 | 3 | 3 | 3 | 2 | 2 | 1 | 2 | 3 | 3 | 2 | 1 | 3 | 3 | 40 |
| SH12 | 3 | 3 | 3 | 3 | 2 | 3 | 3 | 2 | 1 | 2 | 3 | 1 | 3 | 1 | 3 | 1 | 37 |
| RMS2 | 3 | 3 | 3 | 3 | 3 | 3 | 3 | 2 | 1 | 3 | 3 | 2 | 3 | 1 | 2 | 3 | 41 |
| Prelude | 2 | 3 | 1 | 1 | 1 | 3 | 3 | 2 | 1 | 1 | 3 | 3 | 1 | 1 | 2 | 1 | 29 |
| Pre-CZ | - | - | - | 2 | 1 | 3 | 3 | 2 | 1 | 1 | 3 | 2 | 1 | 1 | 3 | 3 | - |

## 4. Discussion

The current study is part of a program that is exploring the ideal yeasts to produce high-quality fruit wines in Guizhou. The study aimed to select the most suitable yeast strain for cili winemaking. Recognizing that wine quality is directly dependent on the appropriate yeast strain, we evaluated the oenological properties of potentially suitable yeast strains, namely the growth curve, fermentation vigor, total $SO_2$ tolerance, flocculation, and $H_2S$ production capacity. In addition, we directly analyzed the resultant wines for alcohol content, residual sugar, titratable acidity, volatile acidity, ascorbic acid content and volatile compounds.

The growth curves of all *Saccharomyces* strains were similar, having a short, if any, lag phase and an exponential phase with a high growth rate, which showed a typical fermentative growth characteristic [38]. The growth curve of the *Torulaspora delbrueckii* Prelude strain showed a prominent lag phase but total yeast growth accelerated after inoculation with *S. cerevisiae* CZ on day 5. Growth of Prelude alone remained relatively slow. The time to the typical stationary phase of all the *Saccharomyces* strains was about eight days while the *Torulaspora* strain took about 15 days to reach that phase. In typical grape wine fermentations, the stationary phase occurs after about 12 days but may occur earlier at warm temperatures.

To promote production of medium chain length fatty acids (MCFAs)—which contain four to 12 carbons—and their ethyl esters, which impart 'fruity' aromas to wines [39], the fermentations in this study were conducted at low temperature, 18 °C [40], low juice solids concentration, <100 NTU [41], and under anaerobic conditions [42], as generally used in white grape winemaking [43]. It is well recognised that each of these conditions restrain yeast activity to the extent that it may become sluggish or 'stuck' [44]. Indeed, in this study, none of the ferments reached the potential alcohol content of *ca* 13% *v/v* and all wines contained residual sugar. For example, the alcohol content of the RMS2 wine (11.55 %vol) was closest to the potential alcohol and yet the residual sugar content was relatively high (58.19 g/L). The well-known sensitivity of *T. delbrueckii* to ethanol [8], together with the effect of MCFA-induced yeast growth inhibition [45] probably led to both the particularly high residual sugar content and its high variability (standard deviation) in that wine.

Although the fermentation conditions favoured MCFA production, those compounds, particularly octanoic and decanoic acids, are known to inhibit yeast fermentation [46,47]. Viegas et al., 1989 [46], reported that in fermentation at 30 °C, octanoic acid, and decanoic acid became inhibitory at concentrations exceeding 16, and eight mg/L, respectively. In our study, about six months after fermentation had terminated, the concentrations of those acids ranged from 17.26 to 40.14, and 0.82 to 16.78 mg/L, respectively. We did not determine MCFAs in the juice before fermentation but Zhou et al., 2011 [48], reported that the octanoic acid content was highest of the five fatty acids detected in cili juice from wild plants in Hubei province. Significantly, Huang et al., 2021 [23], found that the headspace concentrations of octanoic acid of cili juice from five different geographic locations in Guizhou province ranged from 1.32 to 12.71 mg/L: the highest content being in cili grown in Liupanshui, Panzhou. These observations indicate that the initial octanoic content of our juice may have been substantial—particularly as the cili used in our study was grown in Liupanshui, Panzhou—but in any event, the high octanoic acid and decanoic contents we found most certainly inhibited the fermentations.

In this study, we used UHT-preserved cili juice, which was followed by HTST pasteurisation prior to yeast inoculation to guard against microbial contamination. In considering whether those treatments contributed to the incomplete fermentations, we note that various pre-ferment thermovinification treatments are commonly used in white grape winemaking to decrease microbial loads, inactivate oxidative enzymes of fungal origin and inactivate enzymes involved in production of C6 alcohols [45]. However, we also note that HTST or UHT-treated acai juice had greater fatty acid content than non-thermally preserved juice [49], thus indicating that our pasteurization treatments may have preserved octanoic acid originally present in the cili juice. The yeast strains in our study varied in their toler-

ance to total $SO_2$ concentration, which is an important oenological trait for wine yeasts [50]. The range of total $SO_2$ in this study (0-400 mg/L) was greater than the previously reported range (0-300 mg/L) [8]. Most of the yeast strains had more than a 50% relative biomass concentration at 300 mg/L total $SO_2$, which indicated that they could tolerate up to that total $SO_2$ concentration. Notably, the Prelude strain had a relative biomass concentration of more than 60% at 400 mg/L total $SO_2$.

Although there were significant differences in flocculation values, all the yeast strains in this study were highly flocculant and, although not specifically compared, all wines clarified readily. Strains F33 and Prelude were the highest $H_2S$ producers while CZ, RMS2, SH12 produced no $H_2S$. The low $H_2S$ production of X16 was consistent with previous findings [8].

Titratable acid present in wine is primarily important for the perception of sour taste [20]. The titratable acid content of all cili wines, expressed as tartaric acid equivalents, was 10–11 g/L. The most prevalent organic acids in cili fruit, all of which contribute to titratable acidity are ascorbic acid, malic acid, lactic acid, tartaric acid, citric acid, oxalic acid, and succinic acid, of which, ascorbic acid accounts for more than 60% [51]. In this regard, it is notable that the content of ascorbic acid in the cili juice was *ca.* 5.3 g/L.

In amounts greater than *ca* 1.2 g/L volatile acidity is an indicator of microbial spoilage. The volatile acidity of all the cili wines was less than 1.2 g/L, which indicates that there was no microbial spoilage during fermentation.

Volatile compounds are important factors affecting the aroma and flavor quality of wine [52]. Overall, we detected 61 volatile compounds in the headspace of the cili wines. By comparison the number of volatile compounds reported in cili juice are 67 [23], and 37 [53]. Although fermentation is expected to significantly change the profile of volatile compounds, esters, higher alcohols, and volatile phenols, were the predominant constituents of our cili wines and the cili juice examined by Huang et al. 2022 [23].

Higher alcohols are alcohols with more than two carbon atoms. In this study, the range in total content of higher alcohols (117.61–177.23 mg/L) in the wines was considerably higher than that reported by Huang et al., 2022 [23], for cili juices, viz., 5.6–57.4 mg/L. This may be related to fermentation effects and/or differences in juice sources and preparation. Given the considerable differences, further work to identify their source is warranted. Higher alcohols are substrates for acetate ester production [54].

At low concentrations in grape wine, ethyl and acetate esters can increase flavor complexity and enhance the aroma with floral, fruity, and cut grass notes but at high concentration in wine (>200 mg/L total esters) they may become unpleasant and considered a fault. They may also impart a bitter taste and at excessive levels they may even become harmful to human health [55]. In this study, the range in total headspace ester content among the wines was 182.64 (S103)–603.6 (Prelude) mg/L.

The CZ yeast is a *Saccharomyces* strain previously isolated in our laboratory. In this study, all fermentation and wine quality parameters of CZ—except headspace acetaldehyde content—were within the range of the other *Saccharomyces* yeasts. To explore the impact of *Torulaspora delbrueckii* on cili ferments, the commercial strain, Viniflora Prelude, used alone, and in sequential inoculation with CZ, was compared with CZ used alone. Prelude when fermented alone, produced higher headspace concentrations of higher alcohols, acids, esters and other compounds, and lower concentrations of aldehydes and volatile phenols, than when CZ alone was used. However, sequential fermentation with Prelude followed by CZ, produced higher concentrations of all headspace volatiles than CZ alone. Notably, the combination of Prelude sequentially followed by CZ increased the number of volatile compounds identified by only seven compared with CZ alone. Among the seven compounds, the odorless dipeptide DL-alanyl-L-alanine was the only one present in a considerable amount. Similarly, while the headspace of wine fermented with CZ alone had four compounds that were not present in the sequential ferment, none of those compounds were present in amounts likely to affect wine aroma.

Although the wines fermented with Prelude alone, or Pre/CZ, had somewhat greater total headspace concentrations of higher alcohols than those made from CZ alone, those concentrations were less than those produced by five of the other *Saccharomyces* yeasts. Similarly, considering the individual higher alcohols of the Prelude, Pre/CZ and CZ wines, there was generally little difference between the headspace contents of those higher alcohols present in amounts likely to contribute significantly to the aroma of the wines. Furthermore, those higher alcohols, which were lower in the CZ headspace, were present in the headspace of wines fermented with several other *S. cerevisiae* strains, at levels which greatly exceeded those of the Prelude and Pre/CZ wines.

It has been reported that *T. delbrueckii* produces less volatile acidity in grape wine than *S. cerevisiae* [56]. We found that the Prelude strain did produce less volatile acidity—predominantly due to acetic acid—in the cili wine, and less acetic acid in the headspace, than CZ.

Overall, the headspace concentration of esters of the Prelude wine was 1.6 times higher than that of the CZ wine. Furthermore, some of the most influential wine esters, viz., ethyl acetate, isobutyl acetate, 3-methyl butyl acetate, phenethyl acetate and isoamyl acetate were present in the headspace of the Prelude wine at two to three times that of the CZ wine headspace. However, sequential fermentation with Prelude followed by CZ resulted in a smaller, viz., 1.3 times, increase in the overall headspace concentration of esters over that of the CZ wine. Notably, the overall headspace ester concentration of CZ was higher than that of the other *S. cerevisiae* strains. There were several headspace esters that were present in the Prelude wine that were not present in wines fermented with *S. cerevisiae* alone or in combination with Prelude, viz., ethyl propionate, n-propyl acetate, 2-pentylacetate, diethyl carbonate, (Z)-ethyl pentadic-9-enoate, 3-octanoate, pentyl ester, and phenethyl propionate. Most of those esters were at relatively low concentrations, however, they may be indicators of fermentation of cili by the Prelude strain of *T. delbrueckii* alone.

Previously it has been reported that *T. delbrueckii* produces less acetaldehyde than *S cerevisiae* [54]. Our headspace results with Prelude alone accorded with that finding, but notably when the fermentation was completed sequentially by CZ, the headspace acetaldehyde concentration rose almost 25-fold.

Volatile phenols contribute to wine aroma. Overall, the headspace phenol concentration of cili wines fermented with Prelude alone was lower than that of wine fermented with CZ alone. Cili wine sequentially fermented with Prelude and CZ had the highest concentration of headspace phenols. In this regard, the dominant phenols were butylated hydroxytoluene and 2,4-ditert-butylphenol. Although Prelude produced lower headspace concentrations of phenols than CZ, the amount produced by Prelude was lower than that produced by six other *S. cerevisiae* strains.

Regarding the other volatile compounds, a notable finding was the high headspace concentration of DL-alanyl-L-alanine in the Pre-CZ wine despite its non-detection in either wine fermented separately. Although as previously noted, DL-alanyl-L-alanine is odorless and therefore does not contribute to wine aroma, this finding does indicate biosynthetic interaction between the two yeasts. Previous studies indicate that biosynthetic interaction and separate influences of *S. cerevisiae* and *T. delbrueckii* on grape wine composition are dependent on the yeast strain, the time of introduction of the two types and the fermentation conditions [57,58]. Further study of these influences is required to better understand and utilize both yeast types in the production of high quality cili wines. Nevertheless, except for incomplete fermentation of the reducing sugars, most cili wines met analytical criteria of acceptability in terms of alcohol content, titratable acidity, volatile acidity, ascorbic acid content and volatile compounds. Accordingly, this study confirms that most yeast strains tested are suitable for cili winemaking. However, based on our scorecard, strains RMS2 were the most suitable. Among the other yeasts, strains CZ and X16 appeared suitable. Further study of the sensory qualities of cili wines is required to confirm these findings.

## 5. Conclusions

The adaptability of yeast strains to cili juice is a core factor affecting the fermentation of high-quality cili wine. In this regard, the high octanoic acid content of cili fruit appears to exacerbate the risk of arrested fermentation under conditions generally considered favorable for the production and retention of desirable, ethyl esters of medium chain length fatty acids. Further study will be required to determine ideal conditions to avoid arrested fermentations of this fruit.

Meanwhile, based on a multi-factor, unweighted, scorecard compiled from analyses of fermentation performance and generic wine attributes, we conclude that, of the yeast strains we tested, the *Saccharomyces* yeast strains RMS2 is the most suitable for cili winemaking. Additionally, the *Saccharomyces* strains CZ and X16 are more suitable for winemaking than the other, non-ideal strains. However, given the large enhancement of fruity ester content by *Torulaspora delbrueckii* Prelude strain, following further study, revised weighting of individual factors to meet wine specifications are likely to improve the utility of this scorecard approach.

**Supplementary Materials:** The following supporting information can be downloaded at: https://www.mdpi.com/article/10.3390/fermentation8070311/s1, Table S1: Headspace volatile compounds of cili wines fermented by different yeasts.

**Author Contributions:** Conceptualization, Z.-H.Y. and W.J.H.; methodology-cili wine preparation, G.-D.H., X.-Y.H. and J.-H.P.; methodology-cili wine chemical analysis, J.-S.W., L.-R.Y. and X.-Z.L.; methodology-cili wine GC-MS analysis, X.-Y.H.; data collection and analysis, M.-Z.H.; writing-original draft preparation, Z.-H.Y., G.-D.H., X.-Y.H. and J.-H.P.; writing-review and editing, W.J.H. All authors have read and agreed to the published version of the manuscript.

**Funding:** This research was supported by Guizhou Provincial Science and Technology Foundation (No. [2019]1145, [2021]174) and Key Laboratory of Microbial Resources Collection and Preservation, Ministry of Agriculture and Rural Affairs (No. KLMRCP2021-02).

**Institutional Review Board Statement:** Not applicable.

**Informed Consent Statement:** Not applicable.

**Data Availability Statement:** Not applicable.

**Conflicts of Interest:** The authors declare no conflict of interest.

## Abbreviations

| | |
|---|---|
| X16-*Saccharomyces cerevisiae* | X16 (Laffort) |
| F33-*S. cerevisiae* | Actiflore F33, (Laffort) |
| RMS2-*S. cerevisiae* | Actiflore RMS2 (Laffort) |
| SH12-*S. cerevisiae* | SafOEno SH 12 (Fermentis) |
| GV107-*S. cerevisiae* | SafOEno GV107 (Fermentis) |
| S102-*S. cerevisiae* | SafOEno BC S102 (Fermentis) |
| CZ-*S. cerevisiae* | CZ strain (Guizhou Institute of Technology) |
| S103-*S. cerevisiae* var. *bayanus* | SafOEno BC S103 (Fermentis) |
| Prelude-*Torulaspora delbrueckii* | Viniflora Prelude (Chr. Hansen) |
| Pre-CZ | sequential inoculation with Prelude followed by CZ |

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
