# Peer review of "A Comparative Study of Yeasts for Rosa roxburghii Wine Fermentation"

_fermentation, doi:10.3390/fermentation8070311_

Round 1

Reviewer 1 Report

Authors propose an interesting and promising study from the point of view of its application in optimizing the production of this particular wine, but whose results suggest a potential application in the fermentation of other food products. Discussion is complete and well documented and compared with updated bibliography.

Even when this is a complete and, in general, well-designed study, I would like to include some considerations here:

Section 3.1.1

The determination of the survival rate cannot be done by measuring the absorbance at 600nm as this type of measurement does not discriminate between living or dead cells. This determination should be made by counting CFU/mL. Furthermore, the authors do not include any citations to support this method. Please reconsider repeating this trial with another method or removing it from the study. Please reconsider repeating this assay with another method or to remove it. In addition, as table 4 reflects the survival rate, column 1 is unnecessary and should be removed.

Section 3.1.3

Even though this is an accepted method, in my opinion colour determination should be done by a quantitative colorimetric method.

Section 3.2.1

Table 5: The data corresponding to the Prelude strain show, in some cases, standard deviations close to 50%. With these data, it is difficult (if not questionable) to accept these results. More replicates should have been included in order to obtain more reliable data.

Reviewer 2 Report

My comments and questions are as follows:

The manuscript in its present form is not sufficiently convincing in terms of its novelty. What is the significance of making cili wine with commercially available yeasts? Will it be better than traditional, have a different character, be more commercially saleable?

What justifies the use of commercially available wine starter cultures for cili wine production as autochthonous yeast strains involved in spontaneous fermentation?

What characteristics were used to select the Saccharomyces cerevisiae active dry yeast cultures?

The SafOEno BC S103 (Fermentis) culture is S. bayanus, not S. cerevisiae.

From an editorial point of view, the manuscript needs reworking and corrections in several places, the space is missing in numerous places between the words, e.g. in lines 39, 67, 98, 121, and 218.

In line 50, in the part "six strains which most", I suggest "that" instead of "which".

The ability to produce H2S is an important selection factor. Since the authors used commercial yeast cultures, for which this property was likely to have been tested, what was the rationale for demonstrating this under laboratory conditions? For several cultures, these properties are available on the manufacturer's website.

Explanatory notes for tables and figures should not be on separate lines, as in Figures 2 and 3 (the punctuation is missing at the end of the headings for Figure 3 and Table 6). The font size and position of the caption for Table 6 is different from that of the other tables. In Table 6, the decimal places have been moved to the next row in several places, which at first view confuses the reader. Table 6 could be presented as supplementary material because of its size. Section 3.2.2 explains its main content.

The scoring system for the standards in Table 3 is mathematically inaccurate due to overlaps. What I mean, for example, is the "Growth curve": >8-14> (scoore 1), >6-8≥ (score 2), >0-6≥ (score 3); „pH”: >2-3> (score 1), ≥3-4≥ (score 3), >4-5> (score 1). The situation is similar for Fermentation vigor, Volatile acidity, Ascorbic acid, Esters, and Phenols. In the first half of Table 3 - "Yeast traits for winemaking"- indicate the units of measurement (day, mg/L, A) wherever possible.

In line 105, does the author mean 50 mg/L free or total SO2?

The unit of measurement of quantities is not standardised. "mL" is given as "ml" in lines 100, 102, 103, 205 and 207. Please correct these.

Table 4 should be redrafted, as it is misleading at first view because of its format. The data for the control measurement (0 mg/L) should be listed in a separate column, not as data under "Survival rate (%)".

In Figure 4 (a), several curve colours (S103, S102, F33, GV107, SH12) are indistinguishable from each other, please correct them.

In Table 7, the captions have slipped down the lines, which is not at all aesthetically pleasing.

In the paragraph between lines 316 and 326 of the Discussion, the argument is incorrect in several places. The optimum growth temperature for the yeast Saccharomyces cerevisiae is 30-35 oC. Below 20 oC, it is sub-optimal for S. cerevisiae and no longer competitive with other Saccharomyces species in fermentation. The 10-15 oC used for the fermentation of white and rosé wines often leads to prolonged fermentation or stagnation. The 18 oC used by the authors did not seem to favour the growth of the strains studied, and therefore the fermentation was not completed in 15 days, as indicated by the high residual sugar content. This is a more plausible explanation than the possibility that the cili may contain substances that inhibit the growth of yeasts. This should be proven by targeted studies. It also mentions in line 322 that the high residual sugar content could possibly come from non-fermentable sugars. There are studies that have investigated the sugar content of Chestnut rose (Rosa roxburghii tratt) fruit e.g. doi: 10.1007/s13197-017-3023-8; doi: 10.3390/molecules21091204. Based on these, what would you assume about the amount of non-fermentable sugars in the cili wines you produce. In my opinion, the residual sugar content is due more to the sucrose added at the beginning of the experiment than to the non-fermentable sugars. What is your opinion on this?

In the study by Vigentini and colleagues (2017), fermentation vigour was performed in YPD broth at pH 6.8 and not in grape juice. YPD is a much more favourable medium for Saccharomyces cerevisiae, which is why a more efficient CO2 production was observed in that experiment over 48 hours. This literature reference is not relevant in this case (in line 325).

Is sulphur used in the production of cili wines, if so, when and in what quantities? Are there any data on the free and total SO2 content of cili wines? The authors set a free (?) SO2 content of 50 mg/L at the end of the main fermentation and then cooled the samples, so it was no longer relevant for the yeasts to survive. The sulphur tolerance of yeasts is relevant when must is treated with sulphur for microbiological stability before fermentation starts. Why would it be important from a fermentation technology point of view for yeasts to be sulphur tolerant when making cili wine?

I have a similar question regarding flocculation. Was it possible to observe during fermentation or later during maturation that the yeasts flocculated, perhaps forming a denser sediment that helped the wine to settle out of the sediment?

By describing the technological steps in cili winemaking, it would be easier to explain why the yeasts under investigation were tested for the particular characteristics they possess.

What flavours were imparted to the cili wines by the volatile components detected in the highest amounts? With only the description of the compounds, the non-expert reader will not be able to imagine the smell and taste of cili wines. To what degree are the primary flavour components (those derived from fruit) retained and to what extent do the different S. cerevisiae yeasts and technological steps contribute to the formation of secondary flavour components?

In terms of their oenological characteristics as stated by the distributors (Laffort, Fermentis), how did the yeasts tested differ or show similarities in the fermentation of cili wines?

Round 2

Reviewer 2 Report

Thank you for taking my comments into account. In my review, I have endeavoured to ensure that your manuscript is a worthy continuation of the work you have already begun. Often, a few convincing arguments in the introduction are enough to show why the research was worthwhile and necessary. That is why I would suggest that the reply you send me should be included in the manuscript. What I mean: „While domestication and utilisation of Rosa roxburghii (Cili) have been attracted the attention of scholars and government in recent years because of its health-promoting properties (see “Wang et al., Food Funct. 2021, 12(4):1432-1451” and “Li et al., Food Chem. 2022, 366: 130509”), there are very few publications concerning the application of contemporary wine fermentation technologies for the production of ‘Western-style’ wines from this fruit; either in Chinese or English.” 

I would suggest that the criteria ("tolerance of fermentation conditions, low production of off-flavours and development of desirable sensory qualities") on the basis of which the yeast strains used were selected be included in the introduction. For example, in line 80.

For figure 4.b, the explanation is missing. "The letters above the columns indicate significant differences."

In lines 562 and 563, the term 'Saccharomyces' should be in italics.
